# Using Mediation Analysis to Understand How Treatments for Paediatric Pain Work: A Systematic Review and Recommendations for Future Research

**DOI:** 10.3390/children8020147

**Published:** 2021-02-16

**Authors:** Hayley B. Leake, G. Lorimer Moseley, Tasha R. Stanton, Lauren C. Heathcote, Joshua W. Pate, Michael A. Wewege, Hopin Lee

**Affiliations:** 1IIMPACT in Health, University of South Australia, Adelaide, SA 5000, Australia; lorimer.moseley@gmail.com (G.L.M.); Tasha.stanton@unisa.edu.au (T.R.S.); 2Centre for Pain IMPACT, Neuroscience Research Australia, Sydney, NSW 2031, Australia; m.wewege@unsw.edu.au; 3Department of Anaesthesiology, Perioperative and Pain Medicine, Stanford University School of Medicine, Palo Alto, CA 94304, USA; lcheath@stanford.edu; 4Department of Physiotherapy, Graduate School of Health, University of Technology Sydney, Sydney, NSW 2008, Australia; joshua.pate@uts.edu.au; 5Department of Exercise Physiology, School of Medical Sciences, Faculty of Medicine and Health, University of New South Wales, Sydney, NSW 2052, Australia; 6Centre for Statistics in Medicine, Nuffield Department of Orthopaedics, Rheumatology and Musculoskeletal Sciences, University of Oxford, Oxford OX3 7LD, UK; hopin.lee@ndorms.ox.ac.uk; 7School of Medicine and Public Health, University of Newcastle, NSW 2308, Australia

**Keywords:** pain, mediation analysis, paediatric, chronic pain, acute pain

## Abstract

Clinicians have an increasing number of evidence-based interventions to treat pain in youth. Mediation analysis offers a way of investigating how interventions work, by examining the extent to which an intermediate variable, or mediator, explains the effect of an intervention. This systematic review examined studies that used mediation analysis to investigate mechanisms of interventions on pain-relevant outcomes for youth (3–18 years) with acute or chronic pain, and provides recommendations for future mediation research in this field. We searched five electronic databases for clinical trials or observational longitudinal studies that included a comparison group and conducted mediation analyses of interventions on youth and assessed pain outcomes. We found six studies (*N* = 635), which included a total of 53 mediation models examining how interventions affect pain-relevant outcomes for youth. Five studies were secondary analyses of randomized controlled trials of psychological interventions for chronic pain; one was a longitudinal observational study of morphine for acute pain. The pain conditions studied were irritable bowel syndrome, functional abdominal pain, juvenile fibromyalgia, mixed chronic pain, and post-operative pain. Fourteen putative mediators were tested, of which three partially mediated treatment effect; seven did not significantly mediate treatment effect and four had mixed results. Methodological and reporting limitations were common. There are substantial gaps in the field with respect to investigating, and therefore understanding, how paediatric interventions work.

## 1. Introduction

Pain is common in childhood and adolescence [1,2], presenting after an injury or procedure, as a consequence of disease, or without any identifiable cause [3]. Effective interventions for acute and chronic pain in young people are critically needed. Many interventions encompass a variety of single and multi-modal treatments including pharmacological, psychological, and physical interventions. The efficacy of such treatments for children and adolescents with acute pain and chronic has been reviewed (e.g., [4,5,6,7]). Some seem effective, but effect sizes are small. One potential explanation is that small effect sizes may represent wide variability in patient response, which could be addressed by better understanding the underlying mechanisms of treatment effects. For example, acceptance and commitment therapy (ACT) is thought to operate via the proposed mechanism of changing psychological flexibility [8], and cognitive behavioural therapy (CBT) is thought to operate by changing maladaptive cognitions (e.g., pain catastrophizing) [9]. Understanding treatment mechanisms would allow targets for paediatric pain interventions to be informed by empirical evidence, rather than presumptive theories. Clinicians would therefore be able to refine and optimize the effectiveness of interventions by selectively targeting mechanisms known to improve outcomes [10].

Mediation analysis is the most frequently used quantitative method for evaluating the mechanisms of interventions [11]. Mediation analyses answer questions about how or why an intervention works, or does not work, by estimating the extent to which interventions exert their effects on outcomes via mediating variables (i.e., ‘mediators’). For example, in a recent mediation analysis, Kendall, et al. [12] showed that cognitive-behavioural therapy (termed the ‘exposure’) reduced anxiety symptoms (termed the ‘outcome’) in youth through improvements in coping efficacy but not through reductions in anxious self-talk. Mediation analyses targeting how or why an intervention works require longitudinal data because the timing of effects between the treatment, mediator and outcome, needs to be established. These analyses also require comparison groups for the intervention because a causal contrast against the treatment cannot be evaluated without one [11,13]. For the purpose of making causal inferences, mediation analyses are best conducted on randomized designs that eliminate potential confounding of the intervention-mediator effects, but non-randomized designs are acceptable if potential confounders are controlled for.

Mediation analyses that examine potential mediators of interventions for paediatric pain exist, but a systematic evaluation of the field is lacking. Systematic reviews of mediation analyses have examined potential mediators of interventions for adults with musculoskeletal pain [14] and back pain [15], but whether we can generalize these findings to children and adolescents with pain, who are developmentally unique and the interventions are modified accordingly, remains unknown. Some mechanisms are common to paediatric acute and chronic pain conditions; others are not [16]. Little is known if the same principle applies to treatment. This review aimed to systematically identify, summarize, and critically appraise studies that use appropriate mediation analysis to examine potential mediators of the effect of any intervention on pain-relevant outcomes for children (3–12 years) and adolescents (13–18 years) with acute or chronic pain.

## 2. Materials and Methods

### 2.1. Study Design and Registration

This systematic review followed a pre-defined protocol that was prospectively registered on PROSPERO (CRD42020160743) and the Open Science Framework (https://osf.io/pw7yb/ (accessed on 10 June 2020)). Deviations from the pre-registered protocol are explained in the following section (also listed in Appendix A). Reporting is according to the Preferred Reporting Items for Systematic Reviews and Meta-Analyses (PRISMA) guidelines [17].

### 2.2. Search Strategy

#### 2.2.1. Database Search

The following electronic databases were searched from their inception up to 4 January 2021: EMBASE (OvidSP), Medline (OvidSP), Emcare (OvidSP), PsycINFO (OvidSP), and Cochrane Central Register of Controlled Trials (CENTRAL). A MEDLINE search strategy was developed in conjunction with a medical librarian and was adapted for other databases (see Appendix A for complete search strategy). We combined three sets of descriptors to capture: (1) paediatric populations (e.g., child, adolescent), (2) pain conditions, (3) mediation analysis (e.g., causal pathway, indirect effect). To ensure we captured all pain conditions, we included broad search terms for pain, as well as specific terms for chronic pain conditions common to paediatric populations [2] (e.g., irritable bowel syndrome, migraine, fibromyalgia). Specific search terms were altered for individual electronic databases (e.g., MeSH). To provide a comprehensive review of the literature, no time or language limits were applied to any of the databases; however, searches were limited to human participants only. The systematic search was conducted by a single investigator (H.B.L.).

#### 2.2.2. Other Sources

Potentially relevant unpublished literature was captured by contacting authors of abstracts identified in the electronic database search as conference proceedings, dissertations, or on ClinicalTrials.gov. Reference lists of relevant systematic reviews [4,5,6,7] and of all studies for which full text articles were retrieved were manually searched for additional studies.

### 2.3. Eligibility Criteria

To be included, studies had to: (1) be published, peer-reviewed reports; (2) have enrolled a sample of children and adolescents aged 3 to 18 years (referred to hereafter as ‘youth’), with pain at baseline of any condition and duration; (3) have investigated any therapeutic intervention delivered in-person or via technology; (4) be a clinical trial (including randomized and non-randomized designs) or an observational longitudinal study; (5) have included any comparator/control group(s); (6) have conducted mediation analysis to investigate the role of one or more mediator(s) in explaining the pathway from an intervention for paediatric pain to a pain-related outcome; (7) have assessed an outcome from the PedIMMPACT recommendations for paediatric pain core outcome domains for acute (e.g., pain intensity, physical recovery), chronic or recurrent pain (e.g., symptoms, emotional functioning) [18]; (8) have included any number of participants (i.e., no limit on sample size); and (9) be reported in English, Portuguese, Spanish, or German (translators were available for only these languages). Studies were excluded if: (1) the independent variable in the mediation analysis was not the intervention, (2) ≥25% of the sample were adults (to ensure majority of sample were youth, while accommodating differing definitions of developmental stages), or (3) ≥20% of participants did not report any pain at baseline (to account for pain conditions that present episodically).

### 2.4. Study Selection

Studies identified by the systematic search were exported to Endnote (Clarivate Analytics, Philadelphia, PA, USA) and duplicates were removed. The studies were then uploaded to Covidence (Covidence.org) for screening. A two-stage screening process was used to identify relevant studies. In the first stage, titles and abstracts of each of the retrieved studies were independently screened by two of three reviewers (H.B.L., J.W.P., M.A.W.). To account for the possibility that eligible records did not mention mediation analysis (or related terms) in the title or abstract, in stage one the reviewers included records that appeared to fit inclusion criteria 1–5 (see Eligibility Criteria). Studies that were clearly irrelevant were excluded. Full texts of remaining studies were retrieved. In stage two, the full texts of all potentially eligible articles were screened for inclusion by two independent reviewers (H.B.L., M.A.W.) against the full eligibility criteria. At each stage, disagreements were resolved by a third reviewer (H.L.).

### 2.5. Data Extraction

Data were extracted in duplicate by two reviewers (H.B.L., M.A.W.) using a customized, piloted data extraction form. A third reviewer was consulted for unresolved disagreements (H.L.). The following data were extracted: study characteristics (e.g., setting, study design); participants (e.g., number, age, gender, pain condition, pain duration); intervention, mediator, and outcome variables (construct, measurement tool, time of measurement); mediation analysis method; measures taken to control for confounders; testing of moderated mediation paths; effect sizes, point estimates and 95% confidence intervals (CIs) of path *a*, path *b*, total, direct and indirect effects, proportion mediated; and the authors’ conclusion(s). We also considered the type of mediation analysis used, as one of two broad categories [19]—(a) the traditional statistical methods that use Baron and Kenny’s framework, the difference and product of coefficients approach [20] or (b) the modern and flexible counterfactual (or potential outcomes) approaches that use simulation based methods [21].

### 2.6. Outcomes

We included studies that directly assessed, or used a composite outcome assessment that includes, one or more of the recommendations for paediatric pain core outcomes for acute, chronic or recurrent pain (PedIMMPACT) [18]. These outcomes include pain intensity, global judgment of satisfaction with treatment, symptoms and adverse events, physical recovery, physical functioning, emotional response, emotional functioning, economic factors, role functioning, and sleep. It is recommended to use child-report as primary as most children above the age of 8 years can provide a valid self-report [22,23]—if parent, physician, and child reports for the same outcome were provided, and ≥80% of the sample were 8 years or older, only the child-reported outcome was included in the manuscript (the parent- or clinician-report was included in Appendix A).

### 2.7. Quality Assessment

Mediation studies meeting the inclusion criteria were assessed for methodological quality using a bespoke appraisal tool. The bespoke appraisal tool was adapted from recommendations for mediation analyses by Vo, et al. [24], which includes items on planning (e.g., protocol registration, sample size estimation), conduct (e.g., handling missing data, adjustment for confounding variables) and reporting (e.g., use of causal diagram, reporting point estimates and confidence intervals). The recommendations of Vo, et al. [24] were designed for controlled trials (RCTs) and needed to be adapted for use with non-randomized designs that require additional adjustment of confounders. Confounders are variables that could induce a spurious (non-causal) association between intervention (usually called ‘exposure’ in mediation analyses) and mediator (path *a*), mediator and outcome (path *b*), or intervention and outcome (path *c*) (see Figure 1) [15]. When an intervention is randomised (e.g., in an RCT) the intervention-mediator and intervention-outcome effects can be considered unconfounded and adjustment only needs to be made for potential confounders of mediator-outcome effects. However, when the intervention is not randomised (e.g., in non-randomized designs) it is necessary to control for all possible confounders of effects (i.e., path *a*, path *b*, path *c*). Therefore, when the appraisal tool was used to assess the quality of non-randomized designs, we adapted two items (items 1.3 and 2.5) so that adjustment of intervention-mediator and intervention-outcome confounders would be considered. We also operationalised item 3.7 such that this item assessed if studies discussed the validity of the following causal assumptions: temporal ordering and no unmeasured confounding [21]. We acknowledge that the validity of other causal assumptions could have been assessed here (e.g., positivity, consistency) but have not. Two reviewers independently appraised the methodological quality of the studies (H.B.L., M.A.W.); disagreements between the two reviewers were resolved via discussion, or a third reviewer (H.L.). Use of bespoke appraisal tool is a deviation from the protocol, as the recommendations by Vo, et al. [24] were published after protocol pre-registration.

### 2.8. Data Synthesis

We considered youth across developmental stages, including children (3–12 years) and adolescents (13–18 years). Where possible, we aimed to interpret the results according to these subgroups, although we acknowledge that a 17-year-old may indeed be more similar to an adult than to a 13-year-old. In this review, we included any intervention but reported results according to intervention type (e.g., psychological, pharmacological). Heterogeneity in pain condition, intervention, mediators, and outcomes precluded meta-analysis, so data were qualitatively synthesized in accordance with the Synthesis Without Meta-analysis (SWiM) reporting guidelines [25]. Narrative analysis included a description of author(s), year of publication, country, study design, population, sample size, intervention type and duration, outcome measures, and schedule of outcome assessments. Studies were grouped based on duration of pain, either as acute pain (duration less than 3 months) or chronic pain (duration of three months or longer [26]). This grouping was a deviation from our initial protocol and reflects our consideration that the mechanisms of treatment effect may depend on pain duration. Tables are also ordered in relation to clinical population (i.e., similar conditions are presented together).

Where possible, outcomes were reported for each model including point estimate, confidence interval and significance levels of path *a*, path *b*, total effect, direct effect, indirect effect, and proportion mediated. In this review, an indirect effect was reported as significant if the 95% confidence interval did not include 0. To assist with comparison between outcomes, data were transformed to present standardized regression coefficients and 95% CIs where possible. In studies where a 90% confidence interval was reported, this was transformed to a 95% confidence interval using the reported standard error. In studies that reported unstandardized correlation coefficients, these were transformed to standardized coefficients (calculation 2.3 in [27]). In studies that used a product of coefficient approach [20] but did not report a correlation coefficient for the indirect effect, this was calculated by multiplying the coefficients of path *a* and path *b* [11].

## 3. Results

### 3.1. Study Selection

The search strategy identified 7334 records for consideration, of which 187 potentially relevant full-text articles were retrieved and screened to determine eligibility. The main reason for exclusion at full-text screening was due to irrelevant study design; full details of reasons for exclusion are provided in Appendix A. Six articles met the inclusion criteria and were included for review (Figure 2).

### 3.2. Characteristics of the Included Studies

Six included studies (comprising 635 participants) were undertaken in USA (*n* = 3) and Sweden (*n* = 3) and published between 2011 and 2020. Sample sizes at baseline ranged from 30 to 200 participants. One study included only children (i.e., 8–12 years); one study included only adolescents (i.e., 13–17 years); four studies included children and adolescents (e.g., 7–17 years [28]). Most (72.9%) participants were female. Five studies included participants with chronic pain, one study included participants with acute pain. Five studies were secondary analyses of previously published RCTs; one study was a prospective, non-randomized, observational study. All studies used a statistical analytic approach to mediation analyses [20]. Five studies analysed the product of coefficients and one study conducted structural equation modelling. Table 1 provides a summary of the characteristics of each study. Two included studies [28,29] reported results of mediation analyses using both child- and parent-reported outcomes. We describe the results of mediation analyses using child-reported outcomes. Results using parent-reported outcomes are provided in Appendix A.

### 3.3. Methodological Quality

The individual study quality assessments are presented in Table 2, with justification provided in Appendix A. Across six studies, 53 mediation models were tested. All models performed well in a few domains of methodological quality for mediation studies. In all 53 models, the choice of mediator was based on clinical rationale, and the mediator was assessed prior to the outcome. All studies used an appropriate framework for analysis—either the product of coefficient approach (35/53) or structural equation modelling (18/53)—and report their approach.

Frequently, models were inconsistent in their engagement with quality domains. Four mediation models reported that they were prospectively planned. Causal diagrams were reported to outline 23/53 mediation models. One model analysed non-randomized data and controlled for intervention-mediator (path *a*) confounders; an adjustment that was not required for the remaining 52 models that analysed randomized data. However, only 21/53 models adjusted for mediator-outcome (path *b*) confounders. Most models (44/53) included additional analyses to justify the validity of the assumption of temporal ordering. 

In some key domains of quality for mediation analysis, studies performed poorly. No power calculations were planned or reported for any model. Only two of the models included potential interactions (i.e., moderators) in analysis, but none evaluated and reported goodness-of-fit indices. Justification for the plausibility of the assumption of no unmeasured confounding was not provided for any models, and no sensitivity analyses were conducted to test this assumption.

### 3.4. Models and Mediators Tested

Across the six included studies, the number of mediation models tested ranged from 1 to 24, amounting to a total of 53 mediation models (Table 1). The number of mediation models tested per study varied as each study assessed differing numbers of putative mediators, outcomes, and time-points of outcome assessment. For example, Lalouni, et al. [29] tested two models assessing the effect of two putative mediators on one outcome measured at one time point; whereas Wicksell, et al. [30] tested 24 models assessing the effect of six putative mediators on two outcomes measured at two time points (detailed results in Appendix A). Of the 53 mediation models that were tested, 12 models found variables that significantly mediated treatment effects, and 41 models assessed variables that did not mediate treatment effects.

In total, 14 putative mediators were assessed in the 53 mediation models. Three mediators were found to significantly mediate treatment effect (morphine consumption, GI-specific avoidance behaviour, GI-specific anxiety), seven did not significantly mediate treatment effect (perceived stress, solicitousness, pain coping, coping efficacy, self-efficacy, kinesophobia, pain intensity) and four had mixed results (perceived pain threat, pain catastrophizing, pain impairment beliefs, pain reactivity). A summary of results of mediation analysis are provided in Table 3; detailed results are provided in Appendix A.

### 3.5. Mediators of Interventions for Acute Pain

One study analysed data from a non-randomized observational design, that investigated an intervention for acute pain in children and adolescents (10–18 years) after spinal surgery [31]. One mediation model was analysed, that explored the mechanism of peri-operative intravenous (IV) acetaminophen compared to no IV acetaminophen, on hospital length of stay. A statistical mediation analysis using a product of coefficient approach [20] of complete cases (*n* = 114) indicated that 79% of the intervention effect was mediated by a decrease in post-operative morphine consumption (indirect effect = −0.31; Sobel’s test: *p* = 0.013).

### 3.6. Mediators of Interventions for Chronic Pain

Five studies analysed 52 mediation models of interventions for youth with chronic pain. All studies performed secondary analyses of data from RCTs and all investigated mechanisms of psychological interventions. A variety of pain conditions were studied, including gastro-intestinal (GI)-related pain (irritable bowel syndrome [IBS], functional abdominal pain) [28,29,32], and non-GI related pain (mixed chronic pain, fibromyalgia) [30,33]. A summary is provided below, and available detailed results of each model are provided in Appendix A.

#### 3.6.1. GI-Related Chronic Pain

GI-specific avoidance behaviour mediated 67% of the treatment effect of internet-delivered CBT compared to waitlist control (*n* = 101) at reducing GI symptoms for adolescents (13-17 years) with IBS (indirect effect = −0.37 [95% CI, −0.09, −0.62]) [32]. In another study, GI-specific avoidance behaviour also mediated treatment effect of internet-delivered CBT compared to treatment as usual (*n* = 90) at reducing GI symptoms for children (8–12 years) with FAPD (indirect effect = −1.73 [0.48, 3.64]) [29].

GI-specific anxiety mediated the treatment effect of internet-delivered CBT compared to treatment as usual (*n* = 90) at reducing GI symptoms for children (8–12 years) with functional abdominal pain disorder (indirect effect = 2.23 [0.66, 4.37]) [29]. 

Perceived pain threat mediated the intervention effect of social-learning CBT on reducing pain intensity at 3-months (indirect effect = −0.11 [−0.18, −0.03]) and 6-months post-treatment (indirect effect = −0.07 [−0.13, −0.003]) for children and adolescents (7–17 years) with FAPD [28]. However, in the same study, perceived pain threat did not mediate the treatment effect on reducing pain intensity at 12-months; nor did perceived pain threat mediate the treatment effect on reducing GI symptom severity at 3-, 6-, or 12 months [28].

Pain catastrophizing mediated the treatment effect of social-learning CBT at reducing GI symptom severity at 3-months (indirect effect = −0.02 [−0.03, −0.001]). However, in the same study, pain catastrophising did not mediate the treatment effect on reducing GI-symptom severity at 6- or 12- months; nor did pain catastrophizing mediate the treatment effect on reducing pain intensity at 3-, 6-, or 12 months [28].

Perceived stress did not significantly mediate the treatment effect of internet-delivered CBT compared to a waitlist control (*n* = 101) for adolescents (13–17 years) with IBS (indirect effect = 0.002 [95% CI, −0.08, 0.09]) [32].

Parent solicitousness did not significantly mediate the treatment effect of social-learning CBT (i.e., children learning through social observation of parent behaviour) compared to education on either of two outcomes (GI symptom severity, pain intensity) for children and adolescents (7–17 years) with FAPD, at any of the three measured time points (3-, 6-, and 12-months post treatment) [28]. 

Parent-reported outcomes: Perceived pain threat, pain catastrophizing, parent solicitousness, GI-specific avoidance behaviour and GI-specific anxiety were also evaluated as mediators of treatment effect on outcomes that were parent-reported. Findings were largely similar to those of child-reported outcomes (see Appendix A).

#### 3.6.2. Non-GI-Related Chronic Pain

Pain reactivity mediated the treatment effect of acceptance and commitment therapy (ACT) compared to a multidisciplinary treatment approach with amitriptyline (*n* = 30) for children and adolescents (10–18 years) with mixed chronic pain conditions, at reducing pain interference at 3.5-months (indirect effect = 0.03 [unstandardized coefficient = 1.10; 95% CI = 0.08, 3.01]) and 7-months (indirect effect = 0.04 [unstandardized coefficient = 1.69; 95% CI = 0.17, 4.32]) and depression at 3.5 months (indirect effect = 0.04 [unstandardized coefficient = 5.43; 95% CI = 0.01, 14.77]). However, in the same study, pain reactivity did not mediate the treatment effect on reducing depression at 7 months [30].

Pain impairment beliefs mediated the treatment effect of ACT compared to a multidisciplinary treatment approach with amitriptyline (*n* = 30) for children and adolescents (10–18 years) with mixed chronic pain conditions, at reducing depression at 3.5-months (indirect effect = 0.05 [unstandardized coefficient = 1.75; 95% CI = 1.75, 14.59]) and 7-months (indirect effect = 0.07 [unstandardized coefficient = 11.56; 95% CI= 2.46, 26.55]). However, in the same study, pain impairment beliefs did not mediate the treatment effect on reducing pain interference at 3.5- or 7-months [30].

Pain catastrophizing did not mediate the treatment effect of CBT compared to fibromyalgia education (*n* = 100) at reducing either of two outcomes (functional disability, depression) at 6-months post-treatment for children and adolescents (11–18 years) with juvenile fibromyalgia [33]. In another study, pain catastrophizing also did not mediate the treatment effect of ACT compared to a multidisciplinary treatment approach with amitriptyline (*n* = 30) for children and adolescents (10–18 years) with mixed chronic pain conditions, on either of two outcomes (pain interference, depression) at any measured time-point (3.5- and 7-months post-treatment) [30].

Pain coping and coping efficacy did not mediate the treatment effect of CBT compared to fibromyalgia education (*n* = 100) at reducing either of two outcomes (functional disability, depression) at 6 months post-treatment for children and adolescents (11–18 years) with juvenile fibromyalgia [33].

Self-efficacy, kinesiophobia, and pain intensity did not mediate the treatment effect of ACT compared to a multidisciplinary treatment approach with amitriptyline (*n* = 30) for children and adolescents (10–18 years) with mixed chronic pain conditions, on either of two outcomes (pain interference, depression) at any measured time-point (3.5- and 7-months post-treatment) [30].

## 4. Discussion

There were several important findings from this review. First, the vast majority of analyses reveal variables that did not mediate treatment effect, but there is value in identifying mechanisms that do not explain why interventions work. Second, the mediators that were identified in this review suggest promising targets for interventions. Finally, there are meaningful opportunities to move the field forward by addressing methodological and design considerations.

The evidence in this review provides more guidance about mechanisms that do not explain why interventions for youth with pain work, rather than mechanisms that do. That is, 41 of the 53 models did not reveal mediators of treatment effect. While these findings would need to be replicated in a larger number of methodologically-sound studies to increase validity, they provide preliminarily evidence that treatments for youth with pain may not operate via current theoretical explanations. For example, Kashikar-Zuck, et al. [33] found that neither pain catastrophizing, coping efficacy or pain coping mediated improvement in functioning or depressive symptoms following CBT. This indicates that CBT works through other undiscovered mechanisms and identifying such mechanisms would increase the ability to adapt CBT to improve its effectiveness in this population. However, identifying variables that do not mediate treatment effects does not necessarily mean that theoretical explanations for how treatments work are unsupported. Two studies in this review [30,32] assessed mediation models including variables that did not align with hypothesised theories of how interventions work, and other variables that did. For example, Wicksell, et al. [30] assessed six potential mediators for the effect of ACT on depression and pain interference. By demonstrated mediation effects in the variables that aligned with the hypothesised theory of ACT (i.e., pain impairment belief, pain reactivity), but not in the variables that did not (i.e., self-efficacy, kinesiophobia, pain catastrophizing, pain intensity), finding from this study support theoretical explanations for how ACT works.

This review revealed mediators that may be promising targets of interventions for youth with pain. Two studies in this review assessed the outcome of GI-symptoms for children [29] or adolescents [32] with GI-related pain, revealing that GI- specific avoidance behaviour may be an important target of exposure-based internet CBT interventions. However, in other included studies, mediators were tested across different populations, types of pain, for different treatments and outcomes, making comparison difficult, and highlighting the limited state of the evidence. While mediation analyses in paediatric pain are common, research with designs able to identify mediators of interventions are rare. Longitudinal data are often used to conduct mediation analysis to investigate mechanisms of the development of symptoms (e.g., [40,41]), rather than mechanisms of interventions. When mediation analyses are used to investigate interventions, some studies use designs that limit the ability to conduct meaningful mediation analyses, such as longitudinal designs without comparison groups (e.g., [42]). Ideally, mediation analyses would be embedded into clinical trials, as a valuable tool to allow us to move beyond efficacy and investigate how intervention for youth with pain do or do not work. More studies of this kind, for each type of intervention with consistent mediators and outcomes are needed to draw firm conclusions about specific mechanisms of interventions for youth with pain.

### 4.1. Methodological Quality of Included Studies

A key methodological strength of all studies in this review was the establishment of temporality, in that all mediators were measured prior to outcomes. Additionally, three studies conducted sensitivity analyses to test the assumption of temporal precedence. Two studies [29,32] included models that assessed mediators at repeated time points—an approach that allows for investigating a treatment with a gradual change in the mediator and outcome [43]. However, methodological limitations were common. No studies estimated a priori sample sizes for indirect or direct effects, which may reflect the complexities and lack of tools available for these calculations [44,45]. Despite their importance, less than half (21/53) of the mediation models in this review controlled for mediator-outcome confounders. Moreover, controlling for some confounders does not exclude the possibility of bias due to other, unmeasured confounders. Sensitivity analyses are recommended to assess the influence of unmeasured confounders [46], but they were not implemented in any study in this review.

### 4.2. Future Recommendations

In light of the methodological issues raised in this review, and the limited number of included studies, recommendations for future research are provided.

#### 4.2.1. Study Effective and Ineffective Interventions

All included studies used mediation analysis to investigate mechanisms of effective interventions. However, there is also value in studying failed mechanisms of ineffective interventions. Mediation analysis of unsuccessful interventions can help identify where the hypothesized causal mechanisms break down. The results of such studies can then be used to reproduce or refine interventions. That is, if we identify mediators that have a causal effect on key outcomes (i.e., path *b*), but the intervention did not affect the mediator (i.e., path *a*), we may modify the intervention to specifically target the mediator. For example, an RCT [47] revealed that a pain science education program was no more effective at reducing functional disability than usual care for adults with chronic back pain, and a secondary mediation analysis was conducted to answer why the intervention did not work [48]. The mediation analysis found that illness perceptions (intermediate variable) were significantly associated with functional disability (outcome), but the intervention did not significantly influence illness perceptions. The authors conclude that illness perception may be an important target in future treatments. That no studies in this review investigated mechanisms of failed interventions is likely influenced by historical perspectives on mediation whereby a statistically significant treatment effect was required to proceed with mediation analysis [20]. However, recent research has shown this requirement to be unnecessary, as indirect effects can be present in the absence of direct effects [13,38,49]. Therefore, mediation analysis of ineffective treatments is encouraged [10,50,51] and could benefit the paediatric pain field.

#### 4.2.2. Assess Shared Mechanisms across Interventions

Studying the same mediators across different interventions can help to identify if different treatments operate through shared mechanisms. Also, if shared mechanisms that affect multiple outcomes can be identified, then we can tailor interventions to specifically target those mechanisms [52]. For example, mechanistic research into treatments for adults with back pain suggest that different psychological interventions produce similar effects on outcomes through a common set of mediators (e.g., increased self-efficacy, reduced pain catastrophizing) [14,15]—the same may be true of interventions for youth with pain. In this review, it is difficult to draw conclusions on shared mechanisms as only two mediators (pain catastrophizing and GI-specific avoidance behaviour) were assessed across multiple studies. Of these, one study identified pain catastrophizing as a mediator of treatment effect [28], and two did not [30,33]; while both studies identified GI-specific avoidance behaviour as a mediator of treatment effect [29,32]. More mediation analyses in the field will provide better insight into the role of shared mechanisms.

### 4.3. Strengths and Limitations of This Review

There are several strengths to this review, including preregistration (now recommended practice in the pain field [53]), a comprehensive search strategy based on established search terms, inclusion of a wide population of youth (aged 3 to 18) and use of two reviewers to independently screen and evaluate studies and extract data. Finally, a key strength of this review is the focus on formal mediation analyses with appropriate study designs (i.e., longitudinal designs with a comparison group), a choice that is reflected in our search terms.

This review also has limitations. Study heterogeneity meant we were unable to compare the magnitude of mediation effects across studies. Publication bias is also a possibility, but the limited number of included studies precluded formal evaluation of it. All but one study [33] in this review reported evidence in favour of one or more of the mediating pathways that were investigated. It is possible that post hoc mediation analyses were only conducted in studies that showed a significant treatment effect, and that primarily those with statistically significant results were published. It is also possible that our search strategy biased towards selecting studies that reported mediation analysis in the title or abstract, which may be more frequent when mediation analyses are significant. Finally, while we reported which studies conducted sample size calculations (none), we did not assess and exclude studies that were inadequately powered for mediation analysis.

## 5. Conclusions

This review identified six studies investigating 53 mediation models across a range of interventions for youth with pain. Across two studies, the literature on exposure-based internet CBT for youth with GI-specific chronic pain suggests that GI-avoidance behaviour may be an important mediators of treatment effect. However, comparison of mediating effects across other studies is limited by study heterogeneity. Investigating how treatments for paediatric pain do, and do not work, is an emerging field. Improving the methodological rigour of such investigations is important. Appropriate designs for mediation, emphasising causal assumptions such as confounding and assessing mechanisms of effective and ineffective interventions, should lead to stronger quality evidence for underlying causal mechanisms of interventions for paediatric pain, and ultimately more effective interventions.

## Figures and Tables

**Figure 1 children-08-00147-f001:**
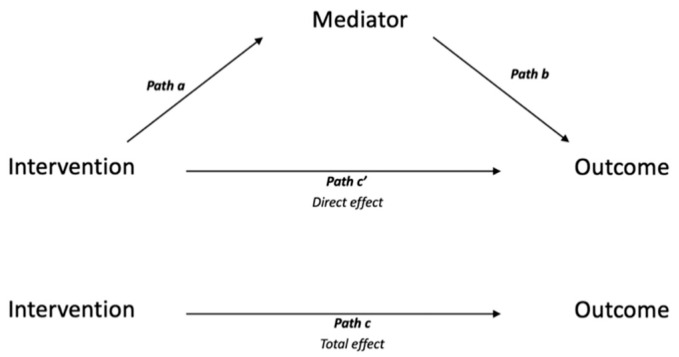
Single mediator model.

**Figure 2 children-08-00147-f002:**
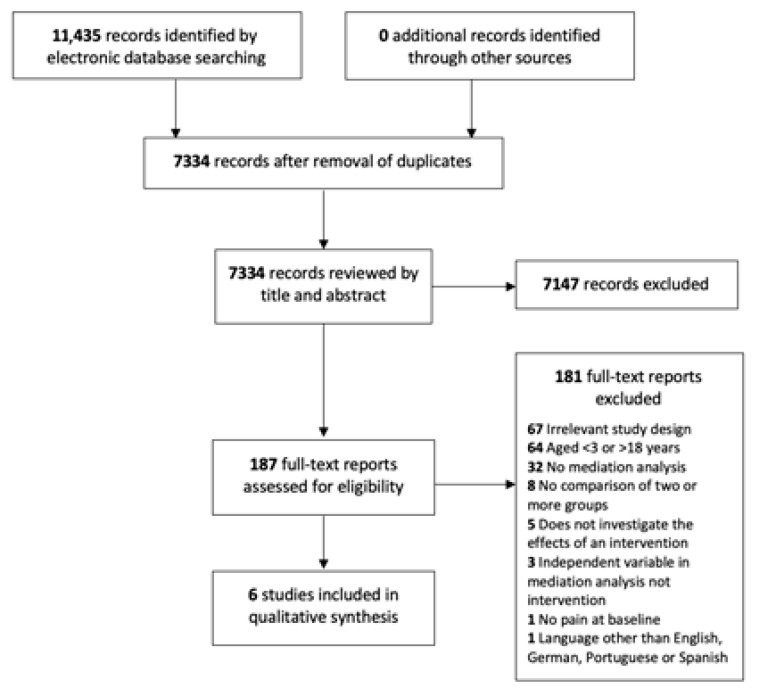
PRISMA flow diagram of the record selection process.

**Table 1 children-08-00147-t001:** Summary of characteristics of included studies.

Study	Design	Sample	Stage of Pain (Baseline)/ Condition	Intervention/ Exposure (*N*)	Length of Intervention (Weeks)	Mediator Variables (Measure)	Outcome Variables (Measure)	Schedule of Assessment (M = Mediator; O = Outcome)	Analysis Method; Single/Multiple Mediator Model; Confounders	Number of Mediation Models Tested
**Acute pain**										
Olbrecht, et al. [31])	Two arm, Observational longitudinal	USA, *n* = 114 [78 female (68%), age 14.4 (SD = 2.0)	Acute pain/ Post posterior spinal fusion for idiopathic scoliosis or kyphosis	Intravenous acetaminophen (*n* = 70) No intravenous acetaminophen (*n* = 44)	<1	Morphine consumption (mg/kg)	Length of stay in hospital (days)	M: post-operatively (varied), while inpatient O: at point of discharge	Product of coefficients [20] with Sobel’s test [34]; Single; Controlled	1
**Chronic pain**										
Bonnert, et al. [32]	Two arm, RCT	Sweden, *n* = 101 [62 female (61%), age 15.5 (SD = 1.6)]	Chronic pain/IBS	Exposure-based Internet-CBT (*n* = 47) Wait-list control (*n* = 54)	10	GI-specific avoidance behavior (IBS-BRQ) * Perceived stress (PSS-10) *	GI symptoms (GSRS-IBS) *	M & O: weekly during treatment	Product of coefficients [20,35] with bootstrapped CIs; Single and multiple; Controlled.	2
Lalouni, et al. [29])	Two arm, RCT	Sweden, *n* = 90 [62 female (69%), age 10.2 (SD = 1.4)]	Chronic pain/Functional abdominal pain	Exposure-based Internet-CBT (*n* = 46) Treatment as usual (*n*= 44)	10	GI-specific avoidance behavior (BRQ-C) * GI-specific anxiety (VSI-C) *	GI-symptoms (PedQL-GI) *†	M: bi-weekly (every 2 weeks) during treatment O: weekly during treatment	Product of coefficients [36,37] with bootstrapped CIs; Single; Uncontrolled.	2
Levy, et al. [28]	Two arm, RCT	USA, *n* = 200 [145 female (73%), age 11.2 (SD = 2.6)]	Chronic pain/Functional abdominal pain	Social learning and CBT (*n* = 100) Education (*n* = 100)	3	Perceived pain threat (PBQ) † Solicitousness (ARCS) † Pain catastrophizing (PRI) *	GI symptom severity (CSI) *† Pain intensity (FPS-R) *†	M: post-treatment O: 3-, 6-, 12-months post-treatment	Structural equation modelling [35,38]; Multiple; Controlled	18
Kashikar-Zuck, et al. [33]	Two arm, RCT	USA, *n* = 100 [93 female (93%), age 15.0 (SD = 1.8)]	Chronic pain/ Juvenile fibromyalgia	CBT (*n* = 50) Fibromyalgia education (*n* = 50)	8	Pain coping (PCQ) * Pain catastrophizing (PCQ) * Coping efficacy (PCQ) *	Functional disability (FDI) * Depression (CDI)*	M: post-treatment O: 6-months post treatment	Product of coefficients [39] with bootstrapped CIs; Multiple; Uncontrolled	6
Wicksell, et al. [30]	Two arm, RCT	Sweden, *n* = 30 [23 female (77%), age 14.7 (range = 10.8–18.1)]	Chronic pain/Mixed chronic pain	ACT (*n* = 15) MDT + amitriptyline (*n* = 15)	10	Pain impairment beliefs (PAIRS) * Pain reactivity (PRS) * Self-efficacy (SES) * Kinesiophobia (TSK) * Pain catastrophizing (PCQ) * Pain intensity (VAS)	Pain interference (PII) * Depression (CES-DC) *	M: post-treatmentO: 3.5-, 7-months post treatment	Product of coefficients [11] with bootstrapped CIs; Single; Uncontrolled	24

* = child report; † = parent report; ACT, Acceptance and Commitment Therapy; ARCS, Adult Responses to Children’s Symptoms; BRQ-C, Irritable Bowel Syndrome Behavioral Responses Questionnaire–Child; CES-DC, Centre for Epidemiological Studies Depression Scale for Children; CDI, Children’s Depression Inventory; CBT, Cognitive Behavioral Therapy; CI, confidence interval; CSI, Children’s Somatization Inventory; FDI, Functional Disability Inventory; FPS-R, Faces Pain Scale–Revised; GSRS-IBS, Gastrointestinal Symptom Rating Scale–Irritable Bowel Syndrome; IBS, Irritable Bowel Syndrome; IBS-BRQ, IBS-specific Behavioral Response Questionnaire; MDT, Multi-Disciplinary Therapy; PBQ, Pain Beliefs Questionnaire; PCQ, Pain Coping Questionnaire; PedQL-GI, Pediatric Quality of Life Gastrointestinal symptoms scale; PPII, Pain Interference Index; PAIRS, Pain and Impairment Relationship Scale; PRI, Pain Response Inventory; PRS, Pain Reactivity Scale; PSS-10, Perceived Stress Scale-10; RCT, Randomized Controlled Trial; SES, Self-Efficacy Scale; TSK, Tampa Scale of Kinesiophobia; VAS, Visual Analogue Scale; VSI-C, Visceral Sensitivity Index for Children.

**Table 2 children-08-00147-t002:** Quality assessment in included studies.

1. Planning	Olbrecht, et al. [31]	Bonnert, et al. [32]	Lalouni, et al. [29]	Levy, et al. [28]	Kashikar-Zuck, et al. [33]	Wicksell, et al. [30]
1.1. Was the mediation analyses planned a priori in the trial protocol?	✘	✓	✓	✘	✘	✘
1.2.1. Was the choice of mediators based on clinical rationale underlying the mechanisms through which the treatment affects the outcome?	✓	✓	✓	✓	✓	✓
1.2.2. Was the choice of mediators based on independent data?	✘	✓	✓	✓	✓	✓
1.3.1. Was there a plan to collect pre- and post-randomization confounders of the exposure-mediator relationship?	✓	n/a	n/a	n/a	n/a	n/a
1.3.2. Was there a plan to collect pre- and post-randomization confounders of the mediator-mediator relationship?	n/a	✘	✘	✘	✘	✘
1.3.3. Was there a plan to collect pre- and post-randomization confounders of the mediator-outcome relationship?	✓	✘	✘	✘	✘	✘
1.4.1. Were the mediators measured prior to the outcome to assure the causal interpretation of the findings?	✓	✓	✓	✓	✓	✓
1.4.2. Was the mediator(s) measured repeatedly?	✘	✓	✓	✘	✘	✘
1.5. Was a causal diagram reported, underlying the causal relationship of the treatment, mediator(s) and outcome?	✓	✓	✓	✓	✘	✘
1.6. Was the sample size for the mediation analysis estimated?	✘	✘	✘	✘	✘	✘
1.7. Was the conduct of a mediation analysis dependent on whether a statistically significant intention-to-treat treatment effect was found?	?	?	?	?	?	?
*‘Planning’ domain summary score*	5/10	6/10	6/10	4/10	3/10	3/10
**2. Conduct**						
2.1.1 Was multiple imputation (or other valid approaches) used to handle missing data?	✓	✓	✓	✓	n/a	✘
2.1.2 If a complete-case analysis was used, did they adjust for baseline covariates that were differentially distributed between responders and non-responders?	✓	n/a	n/a	n/a	n/a	?
2.1.3 Was a sensitivity analysis conducted to assess the impact of different approaches on the findings?	✘	✘	✘	✘	n/a	✘
2.2 Does the study report separate analyses for separate mediators?	n/a	✓	✓	✓	✓	✓
2.3 Does the study use an appropriate framework for analysis?	✓	✓	✓	✓	✓	✓
2.4.1 Does the study evaluate the goodness-of-fit of each model?	✘	✘	✘	✘	✘	✘
2.4.2 Does the study assess potential interaction(s) between treatment and confounding factors, treatment and mediator, mediator and mediator in the mediator and outcome models?	✘	✘	✘	✘	✘	✘
2.5.1a Does the study adjust for exposure-mediator and confounders?	✓	n/a	n/a	n/a	n/a	n/a
2.5.2 Does the study adjust for mediator-mediator confounders?	n/a	✘	✘	✘	✘	✘
2.5.3 Does the study adjust for mediator-outcome confounders?	✓	✓	✘	✓	✘	✘
2.6.1 Does the study perform sensitivity analysis to assess sensitivity of the results to the assumption of no measured mediator-mediator or mediator-outcome confounder?	✘	✘	✘	✘	✘	✘
2.6.2 Does the study perform sensitivity analysis to assess sensitivity of the results to potential measurement errors of the mediators?	✘	✘	✘	✘	✘	✘
2.7 Does the study use apt strategies when some of the mediator-mediator or mediator-outcome confounders are potentially affected by the treatment (e.g., by considering confounders as mediators themselves)?	✘	✘	✘	✘	✘	✘
*‘Conduct’ domain summary score*	5/11	4/11	3/11	4/11	2/9	2/12
**3. Reporting**						
3.1.1. Does the study report the approaches used for mediation analysis?	✓	✓	✓	✓	✓	✓
3.1.2. Does the study provide a causal diagram that underlies the analysis?	✓	✓	✓	✓	✘	✘
3.2.1. Does the study report the sample size calculation?	✘	✘	✘	✘	✘	✘
3.2.2. Does the study report the actual sample size of the mediation analysis?	✓	✓	✓	✓	✓	✓
3.2.3. Does the study report how missing data is handled?	✓	✓	✓	✓	n/a	✘
3.3. Does the study report all confounders considered and adjusted for in the analysis?	✓	✘	✘	✓	✘	✘
3.4.1. Does the study report the model building procedure and the final form of all models used in the analysis?	✘	✘	✘	✘	✘	✘
3.4.2. Does the study report the goodness-of-fit of the models?	✘	✘	✘	✘	✘	✘
3.5. Does the study report the point estimates and the confidence intervals of the different direct, indirect and total treatment effects?	✘	✘	✘	✘	✘	✘
3.6. Does the study report the methods and results of all sensitivity and other additional analyses (in the main paper or appendices)?	n/a	✓	n/a	n/a	n/a	✓
3.7. Does the study discuss the validity of all causal assumptions underlying the analysis (in the main paper or appendices)?	✘	✘	✓	✘	✘	✘
*‘Reporting’ domain summary score*	5/10	5/11	5/10	5/10	2/9	3/11

**Table 3 children-08-00147-t003:** Summary of mediation models assessed in included studies.

Study	Intervention vs. Comparator	Path *a* (I → M)	Mediator	Path *b* (M → O)	Outcome (Child-Reported)	Indirect Effect
Olbrecht, et al. [31]	Intravenous acetaminophen vs no intravenous acetaminophen	+	Morphine consumption	+	Hospital length of stay	+
Bonnert, et al. [32]	Exposure-based internet-CBT vs. waitlist	+	GI-specific avoidance behaviour	+	GI symptoms	+
		–	Perceived stress	+	GI symptoms	–
Lalouni, et al. [29]	Exposure-based internet-CBT vs treatment as usual	+	GI-specific avoidance behaviour	+	GI symptoms	+
		+	GI-specific anxiety	+	GI symptoms	+
Levy, et al. [28]	SLCBT vs. education	+	Perceived pain threat	–	GI symptom severity at 3 months	–
		+	Perceived pain threat	–	GI symptom severity at 6 months	–
		+	Perceived pain threat	–	GI symptom severity at 12 months	–
		+	Perceived pain threat	+	Pain intensity at 3 months	+
		+	Perceived pain threat	+	Pain intensity at 6 months	+
		+	Perceived pain threat	–	Pain intensity at 12 months	–
		+	Solicitousness	–	GI symptom severity at 3 months	–
		+	Solicitousness	–	GI symptom severity at 6 months	–
		+	Solicitousness	–	GI symptom severity at 12 months	–
		+	Solicitousness	–	Pain intensity at 3 months	–
		+	Solicitousness	–	Pain intensity at 6 months	–
		+	Solicitousness	–	Pain intensity at 12 months	–
		+	Pain catastrophizing	+	GI symptom severity at 3 months	+
		+	Pain catastrophizing	+	GI symptom severity at 6 months	–
		+	Pain catastrophizing	–	GI symptom severity at 12 months	–
		–	Pain catastrophizing	–	Pain intensity at 3 months	–
		–	Pain catastrophizing	+	Pain intensity at 6 months	–
		–	Pain catastrophizing	–	Pain intensity at 12 months	–
Kashikar-Zuck, et al. [33]	CBT vs. fibromyalgia education	NR	Pain coping	NR	Functional disability at 6 months	–
		NR	Pain coping	NR	Depression at 6 months	–
		NR	Pain catastrophizing	NR	Functional disability at 6 months	–
		NR	Pain catastrophizing	NR	Depression at 6 months	–
		NR	Coping efficacy	NR	Functional disability at 6 months	–
		NR	Coping efficacy	NR	Depression at 6 months	–
Wicksell, et al. [30]	ACT vs. MDT + amitriptyline	+	Pain impairment beliefs	–	Pain interference at 3.5 months	–
		+	Pain impairment beliefs	–	Pain interference at 7 months	–
		+	Pain impairment beliefs	+	Depression at 3.5 months	+
		+	Pain impairment beliefs	+	Depression at 7 months	+
		+	Pain reactivity	+	Pain interference at 3.5 months	+
		+	Pain reactivity	+	Pain interference at 7 months	+
		+	Pain reactivity	+	Depression at 3.5 months	+
		–	Pain reactivity	+	Depression at 7 months	–
		–	Self-efficacy	–	Pain interference at 3.5 months	–
		–	Self-efficacy	–	Pain interference at 7 months	–
		–	Self-efficacy	–	Depression at 3.5 months	–
		–	Self-efficacy	–	Depression at 7 months	–
		–	Kinesiophobia	–	Pain interference at 3.5 months	––
		–	Kinesiophobia	–	Pain interference at 7 months	–
		–	Kinesiophobia	–	Depression at 3.5 months	–
		–	Kinesiophobia	–	Depression at 7 months	–
		–	Pain catastrophizing	–	Pain interference at 3.5 months	–
		–	Pain catastrophizing	–	Pain interference at 7 months	–
		–	Pain catastrophizing	–	Depression at 3.5 months	–
		–	Pain catastrophizing	–	Depression at 7 months	–
		–	Pain intensity	–	Pain interference at 3.5 months	–
		–	Pain intensity	–	Pain interference at 7 months	–
		–	Pain intensity	–	Depression at 3.5 months	–
		–	Pain intensity	–	Depression at 7 months	–

Statistical significance is defined as 95% confidence intervals that do not contain zero. Mediation models with a statistically significant indirect effect are shaded in grey. (+): Statistically significant association; (−): Statistically non-significant association; (NR): Not reported; ACT, Acceptance and Commitment Therapy; CBT, Cognitive Behavioral Therapy; E, Exposure; GI, Gastrointestinal; M, Mediator; MDT, Multidisciplinary therapy; O, Outcome; SLCBT, Social-Learning Cognitive Behavioral Therapy.

## Data Availability

All data for this study are included in the manuscript and/or Appendix A.

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
