# Peer review of "Using Mediation Analysis to Understand How Treatments for Paediatric Pain Work: A Systematic Review and Recommendations for Future Research"

_children, 2021, doi:10.3390/children8020147_

Round 1

Reviewer 1 Report

Thank you for the opportunity to review this manuscript. The current systematic review is thoughtfully designed and methodologically rigorous. The manuscript is clear and adds to the current literature. I thank the authors for their attention to existent guidelines and offer a few suggestions for improving the manuscript below. 

Methods:

  • Please justify exclusion criteria of > 25% adults and > 20% of participants did not report pain at baseline.

Results:

  • If parent and/or physician report data is available for some studies and analyses, please include this information as well. While I agree child-report should be utilized first, parent-report and physician-report also adds information and is valuable information regarding child functioning.
  • The main results section on Page 7, section 3.6 is difficult to follow and does not provide digestible result to readers. Suggest authors note subsections of GI-related pain and non-GI related pain, then organize results within these subsections by primary mediation domain (e.g., perceived stress, pain catastrophizing), rather than organizing by individual study.

Tables:

  • In black and white print, the reason for shading Table 2 is unclear and makes Table 2 more difficult to interpret. Please reconsider/edit shading on this Table.
  • The reason for shading in Table 3 (i.e., positive indirect results) needs to be defined in the legend.
  • Can a summary score (e.g., %) be provided for the quality assessment of studies in Table 2? Subdomain and Total summary % score would be helpful to interpret this table.

Minor:

  • Page 2, line 46 to 47; This is awkward wording that I didn’t understand and suggest editing.
  • Page 2, line 65; “Mediational analyses are best conducted on randomized designs..”; This is true in the current study context but mediational analyses are helpful examining relationships in other contexts as well (e.g., cross-sectional, longitudinal repeated measures). Suggest tempering this sentence slightly.
  • Suggest giving brief example of PedIMMPACT core outcomes on Page 3, line 120-121.

Reviewer 2 Report

Thank you for the opportunity to review this manuscript. This systematic review evaluates an important topic of understanding treatment mechanisms for pediatric pain interventions. This review will likely be a value add to the field to help move the literature base forward. A few suggestions are offered below for consideration to strengthen the review’s contribution.

Introduction

  • The rationale for the review nicely describes the need to better understand how pediatric pain treatments work
  • It may be helpful to include some examples of possible or theorized mechanisms

Method

  • Inclusion of any type of intervention (i.e., pharmacological, psychological, physical) seems really broad. Would you expect pharmacological interventions to have the same mechanisms of action as psychological interventions? It will be helpful to clarify the focus on any type of intervention for this review and the implications that may have on the results that can be drawn

Discussion

  • Based on the limited number of studies included in the review (n=6), it may be premature to conclusively state that this review provides evidence about mechanisms that do not explain why pain interventions work for youth, rather than mechanisms that do. These conclusions would need replication among a larger number of studies that address the methodological and design considerations identified.

Round 2

Reviewer 1 Report

Thank you for the opportunity to review this resubmission. The authors have addressed my initial concerns and suggested edits. I have no further suggested edits at this time.